# Controlling photophysical properties of ultrasmall conjugated polymer nanoparticles through polymer chain packing

Hubert Piwoński[1], Tsuyoshi Michinobu[2] & Satoshi Habuchi[1]

Applications of conjugated polymer nanoparticles (Pdots) for imaging and sensing depend on their size, fluorescence brightness and intraparticle energy transfer. The molecular design of conjugated polymers (CPs) has been the main focus of the development of Pdots. Here we demonstrate that proper control of the physical interactions between the chains is as critical as the molecular design. The unique design of twisted CPs and fine-tuning of the reprecipitation conditions allow us to fabricate ultrasmall (3.0–4.5 nm) Pdots with excellent photostability. Extensive photophysical and structural characterization reveals the essential role played by the packing of the polymer chains in the particles in the intraparticle spatial alignment of the emitting sites, which regulate the fluorescence brightness and the intraparticle energy migration efficiency. Our findings enhance understanding of the relationship between chain interactions and the photophysical properties of CP nanomaterials, providing a framework for designing and fabricating functional Pdots for imaging applications.

[1] King Abdullah University of Science and Technology (KAUST), Biological and Environmental Sciences and Engineering Division, Thuwal 23955-6900, Saudi Arabia. [2] Tokyo Institute of Technology, Department of Organic and Polymeric Materials, 2-12-1 O-okayama, Meguro-ku, Tokyo 152-8552, Japan. Correspondence and requests for materials should be addressed to S.H. (email: Satoshi.Habuchi@kaust.edu.sa).

New fluorescence microscopy methods often require fluorophores with specific spectroscopic properties. The development of new fluorophores is thus increasingly important. Various types of fluorophores, including organic dyes, fluorescent proteins and semiconductor quantum dots (QDs), have been used in fluorescence microscopy. Conjugated polymer nanoparticles (Pdots) have recently emerged as a new class of fluorescent nanoparticles because their fluorescent properties can be fine-tuned by an appropriate molecular design of conjugated polymers (CPs)[1,2]. Given their superior fluorescent brightness and photostability as well as low cytotoxicity, the applicability of Pdots as fluorescent tags[3,4] and sensors[5–7] has been expanding significantly during the last few years. Recent studies have demonstrated that, in some cases, the performance of Pdots as fluorescent tags surpasses that of conventional organic dyes and QDs[8,9].

Several key factors, including the fluorescence wavelength, brightness of the fluorescence and size of the particle, greatly influence the applications of Pdots. Most bioimaging applications require fluorescence in far-red to near-infrared spectral region[3]. The bright fluorescence with high photostability always provide better quality of fluorescence images[1]. The vast majority of the previously reported Pdots were in the 10–100 nm diameter range[2]. However, when the Pdots are used as fluorescent tags, they should be as small as possible. Especially in clinical applications, the particle sizes should be smaller than 5.5 nm in diameter for complete elimination from the body by renal filtration[2]. One of the unique features of CP materials is their nanoscale intra- and inter-chain excitation energy transfer (EET)[10,11]. EET often causes quenching of fluorescence in CP materials at a length scale of a few nanometres up to tens of nanometres due to efficient funnelling of the excitation energy to the trap sites[12–14]. Given the sizes of the particles, fluorescence brightness of Pdots is highly related to EET. In addition, CP-based fluorescence sensing utilizes EET-mediated fluorescence quenching to enhance detection sensitivity[15]. Clearly, EET is one of the key factors that determines the applicability of Pdots.

Single-molecule spectroscopy studies on single CP molecules and CP aggregates have demonstrated that the conformational state of the polymer chain has a significant effect on their photophysical properties[11,16,17]. Although those studies implied that controlling the physical interaction between the polymer chains in the particles would be important to regulating the EET process and thus controlling the fluorescence properties of Pdots, including their brightness, little is known about the impact of intraparticle chain conformation and chain interactions on the fluorescence properties of Pdots.

Here we report the fabrication of far-red-emitting ultrasmall (diameter ≈ 3.0–4.5 nm) Pdots with high photostability (photobleaching yield ≈ $1.7$–$3.3 \times 10^{-11}$) from an appropriate molecular design combined with proper control of the physical interactions between the polymer chains.

## Results

**Fabrication of the Pdots**. In this study, we fabricated Pdots based on a donor-acceptor-type CPs from a poly(1,8-carbazole)-benzothiadiazole copolymer (PCzBT) and a poly(1,8-carbazole)-dithienylbenzothiadiazole copolymer (PCzDTBT) (Fig. 1a,b). PCzBT and PCzDTBT in tetrahydrofuran (THF) exhibited broad absorption spectra with maxima at 482 and 494 nm, respectively (Fig. 1c,d)[18]. The bands are attributed to intramolecular charge-transfer (CT) absorption between donor (carbazole (Cz)) and acceptor (benzothiadiazole (BT) or dithienylbenzothiadiazole (DTBT)) moieties. Unlike most

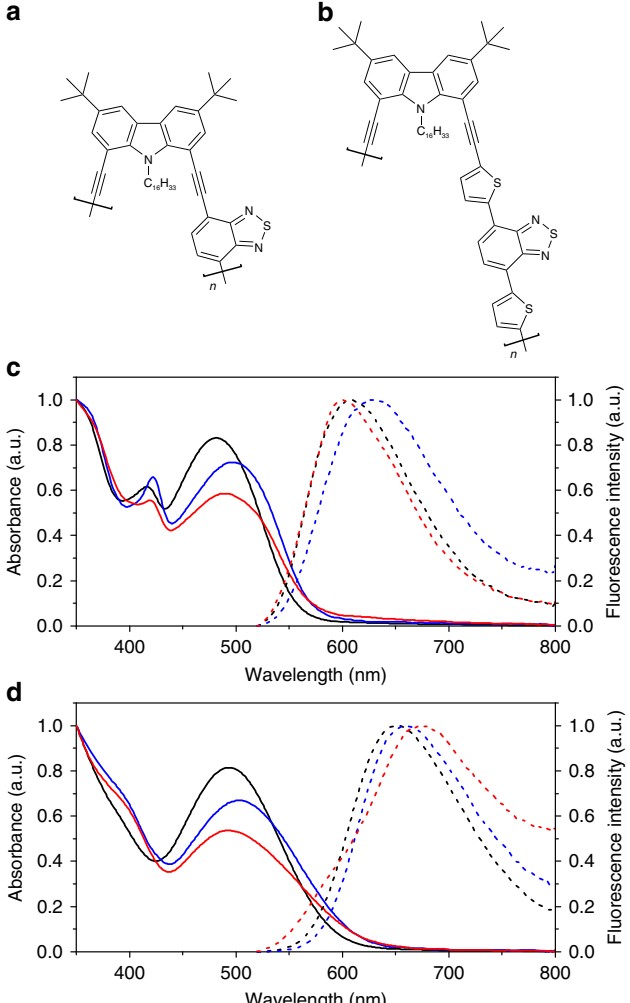

**Figure 1 | Spectroscopic properties of the fabricated Pdots.** Chemical structure of (**a**) PCzBT and (**b**) PCzDTBT. (**c**) Steady-state absorption (solid lines) and fluorescence (broken lines) of PCzBT in THF (black lines) and PCzBT Pdots fabricated at 277 K (PD1-L) (blue lines) and at 343 K (PD1-H) (red lines) dispersed in water. (**d**) Steady-state absorption (solid lines) and fluorescence (broken lines) of PCzDTBT in THF (black lines) and PCzDTBT Pdots fabricated at 277 K (PD2-L) (blue lines) and at 343 K (PD2-H) (red lines) dispersed in water. The fluorescence spectra were recorded upon 510-nm excitation.

CT-type CPs that have a linear/planar shape, our CPs have a nonplanar twisted conformation due to the steric hindrance between the monomer units[19]. PCzBT and PCzDTBT in THF displayed fluorescence from the CT excited states with maxima at 608 and 652 nm, respectively (Fig. 1c,d). The fluorescence quantum yield ($\phi_{\mathrm{fl}}$) and the mean fluorescence lifetime ($\tau_{\mathrm{fl}}$) in THF were respectively determined to be 0.25 and 3.36 ns for PCzBT and 0.50 and 4.63 ns for PCzDTBT (Supplementary Fig. 1, Supplementary Table 1).

The Pdots were fabricated using a modified reprecipitation method (see Methods for details)[20]. A highly diluted THF solution of the CPs was rapidly added to excess water under vigorous ultrasonication at either 277 or 343 K, which was followed by an evaporation of THF. Transmission electron microscopy (TEM) images of Pdots obtained from PCzBT (PD1-L) and PCzDTBT (PD2-L) at 277 K (Fig. 2a,f) showed that they were nearly spherical in shape with the mean diameters of 3.0 and 4.5 nm, respectively. The TEM images also

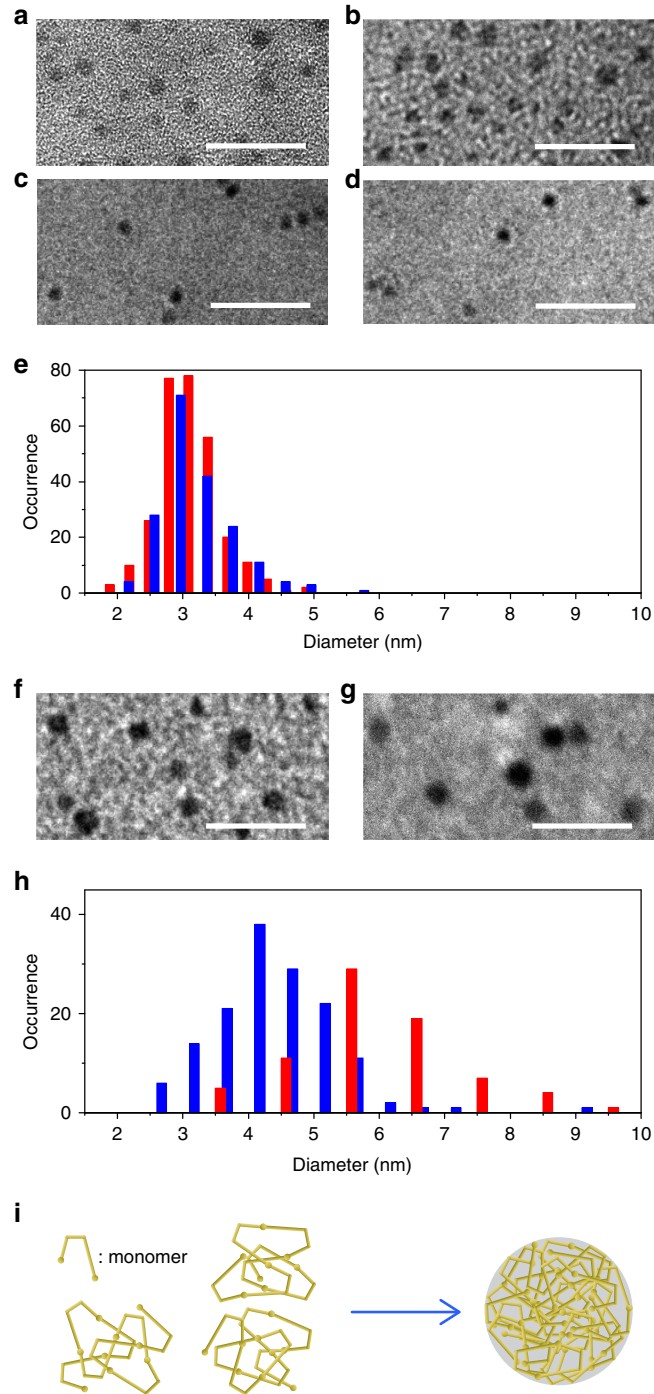

**Figure 2 | Size distribution of the fabricated Pdots.** Transmission electron microscopy (TEM) images of (**a**) PD1-L and (**b**) PD1-H. Cryo-electron microscopy (EM) images of (**c**) PD1-L and (**d**) PD1-H. (**e**) Frequency histogram of the size distribution of PD1-L (blue) and PD1-H (red). TEM images of (**f**) PD2-L and (**g**) PD2-H. (**h**) Frequency histogram of the size distribution of PD2-L (blue) and PD2-H (red). (**i**) Schematic illustration of the conformational state of PCzBT and PCzDTBT in the solution and in the Pdots. Scale bars = 20 nm.

demonstrated narrow size distributions of the obtained Pdots (standard deviation of 0.56 and 1.36 nm for PD1-L and PD2-L, respectively) (Fig. 2e,h). Cryo-electron microscopy images also confirmed the 3.0-nm size of the fabricated Pdots in an aqueous

environment (Fig. 2c). The ultrasmall particle sizes are likely to be due to the nonplanar shapes of PCzBT and PCzDTBT (Fig. 2i, see Discussion for details). PD1-L and PD2-L in water had highly negative zeta potentials of, respectively, −54 and −51 mV at neutral pH, indicating high dispersion stability. The colloidal solution remained stable in the dispersion for a period of at least six months.

Pdots smaller than 10 nm in diameter have been reported by several groups. Wu et al.[21] reported polyfluorene (PF) derivative-based Pdots with 3–7-nm diameters. However, the size was determined by atomic force microscopy (AFM), which usually determines particle sizes to be much smaller than those measured by TEM[22]. Zeigler et al.[23] reported PF-BT copolymer (PFBT)-based Pdots with 7.5-nm diameter (as determined by dynamic light scattering). Only 2% of the total mass was PFBT CPs in the Pdots (that is, on average one PFBT chain per Pdot), and thus the brightness of the Pdots was similar to that of semiconductor QDs of similar size. Also, the zeta potential of the fabricated Pdots was −28 mV, indicating relatively low colloidal stability. Hashim et al.[24] fabricated the smallest Pdots reported so far using polyphenylenevinylene, polyphenyleneethynylene and PFBT derivatives, with the diameters in the range of 3–5 nm. The yield of this preparation seemed to be very low[2]. Pu et al.[25] reported a 3.6-nm conjugated oligoelectrolyte that falls into a different category of conjugated polymer nanoparticles (single-molecule nanoparticles). To the best of our knowledge, the Pdots fabricated from PCzBT (PD1-L and PD1-H, 3 nm in diameter) in this study are among the smallest conjugated polymer nanoparticles reported so far.

**Spectroscopic properties of the Pdots.** The absorption and fluorescence spectra of PD1-L and PD2-L in water exhibited slight red shifts in comparison with those of PCzBT and PCzDTBT in THF (Fig. 1c,d). While the fluorescence quantum yields of PD1-L and PD2-L in water ($\phi_{fl} = 0.16$ and 0.20, respectively) are smaller than those of PCzBT and PCzDTBT in THF (Supplementary Table 1), a much larger reduction in $\phi_{fl}$ is usually observed in far-red emitting Pdots[1,26]. Single particle fluorescence measurements demonstrated that the fluorescence brightness of the obtained Pdots was 5–9 times higher than that of CdSe/ZnS QDs, which emit photoluminescence in the same wavelength range under identical measurement conditions (Supplementary Figs 2–4, Supplementary Table 2). Given the size differences between the particles, PD1-L and PD2-L have more than an order of magnitude higher fluorescence brightness per unit volume compared with those QDs (Supplementary Table 2). This is mainly due to the extremely large molar extinction coefficient ($\varepsilon$) of the Pdots ($\varepsilon \approx 2.8 \times 10^6$ and $1.2 \times 10^7 \, M^{-1} \, cm^{-1}$ for PD1-L and PD2-L, respectively; see Supplementary Fig. 5, Supplementary Table 1), suggesting that CP chains are densely packed in the particles (see Supplementary Note 1). Fluorescence intensity trajectories obtained from single Pdots exhibited nearly constant intensities without any sign of blinking including triplet-state blinking (Supplementary Fig. 6). This is due to the presence of multiple emitters inside the particles that fluorescence independently (see below). The photobleaching quantum yields ($\phi_{bl}$) of PD1-L and PD2-L were estimated to be $3.3 \times 10^{-11}$ and $1.7 \times 10^{-11}$, respectively (Supplementary Fig. 7). These findings demonstrate the successful fabrication of one of the smallest Pdots with the highest photostability reported so far in a reliable manner.

Each PCzBT and PCzDTBT chain may adopt different conformational states because of the relatively short chain lengths (on average 7.9 and 3.3 monomers, respectively), and thus may exhibit different fluorescence properties. Indeed, the

single-chain experiments revealed the large heterogeneity in the fluorescence properties between the chains[19]. However, the single-particle experiments showed the narrow distributions of the fluorescence brightness (Supplementary Fig. 4, Supplementary Table 3) and lifetime (Supplementary Fig. 8, Supplementary Table 3), demonstrating the homogeneous fluorescence properties of the fabricated Pdots. This is most likely due to the large number of the CP chains in each Pdot (see Supplementary Note 1).

During optimization of the fabrication protocol, we found that while the reprecipitation temperature did not affect the size and shape of the Pdots (Fig. 2b,d,e,g,h), spectroscopic properties of the Pdots depend significantly on the temperature during ultrasonication. The PCzBT and PCzDTBT Pdots fabricated at 343 K (PD1-H and PD2-H) exhibited smaller CT absorption (Fig. 1c,d) and lower fluorescence quantum yields ($\phi_{fl} = 0.08$ and 0.09, respectively) compared with the Pdots fabricated at 277 K ($\phi_{fl} = 0.16$ and 0.20 for PD1-L and PD2-L, respectively) (Supplementary Table 1). The oscillator strength (that is, absorption intensity, $f$) of the CT absorption band depends on the rotational twist angle between the Cz and BT moiety (that is, smaller $f$ with larger twist angle)[19], suggesting larger twist angle between the Cz and BT moiety in PD1-H than in PD1-L. Given the structural similarity of PCzBT and PCzDTBT, the difference in the absorption spectra observed in PD2-L and PD2-H can also be interpreted by twist-angle-dependent $f$. A drop in $\phi_{fl}$ is often observed in CP aggregates including Pdots due to the generation of energy trap sites to which the excitation energy is funnelled through EET processes[11,16]. However, no reduction in $\phi_{fl}$ was observed in nanoparticles obtained from a dimeric form of CzBT (Supplementary Fig. 9) under the same reprecipitation conditions (data not shown). This excludes simple aggregation-induced fluorescence quenching as the mechanism of the reprecipitation condition-dependent $\phi_{fl}$ of the Pdots. Rather, this observation implies critical roles of the intraparticle chain conformation and chain interactions in the brightness of the fluorescence.

**Photophysical interaction between the emitting sites**. We next investigated the spatial orientation of the emitting sites in single Pdots using single-particle excitation polarization modulation (Fig. 3a). PD1-L and PD2-L showed greater modulation depths (mean depth = 0.71 and 0.36, respectively) compared with PD1-H and PD2-H (mean depth = 0.43 and 0.19, respectively) (Fig. 3b,c). A large modulation depth in multichromophoric systems corresponds to spatially well-ordered orientation of the absorption transition dipoles of the chromophores[27]. Our results, therefore, demonstrate that the emitting sites in PD1-L and PD2-L are more spatially ordered than those in PD1-H and PD2-H (Fig. 3d).

The spatial order of the fluorophores in multichromophoric systems often affects the efficiency of EET between them. Intra- and inter-chain EET in spatially isolated CP chains or CP nanomaterials (such as CP aggregates) are described by either the strong electronic coupling between the subunits (that is, delocalization of excited state over multiple subunits) or weak coupling (that is, localized excited-state hops between the subunits through Förster-type dipole–dipole interactions)[16,28,29]. The absorption and fluorescence spectra of PCzBT in organic solvents are similar to the dimeric form of CzBT, suggesting that the delocalization of the excited state occurs in up to two monomer units (that is, the dimeric form of CzBT can be treated as the spectroscopic unit of PCzBT)[19]. The fluorescence spectrum obtained for PD1-L in water has a peak wavelength that is similar to that of PCzBT in THF (Fig. 1c). In addition,

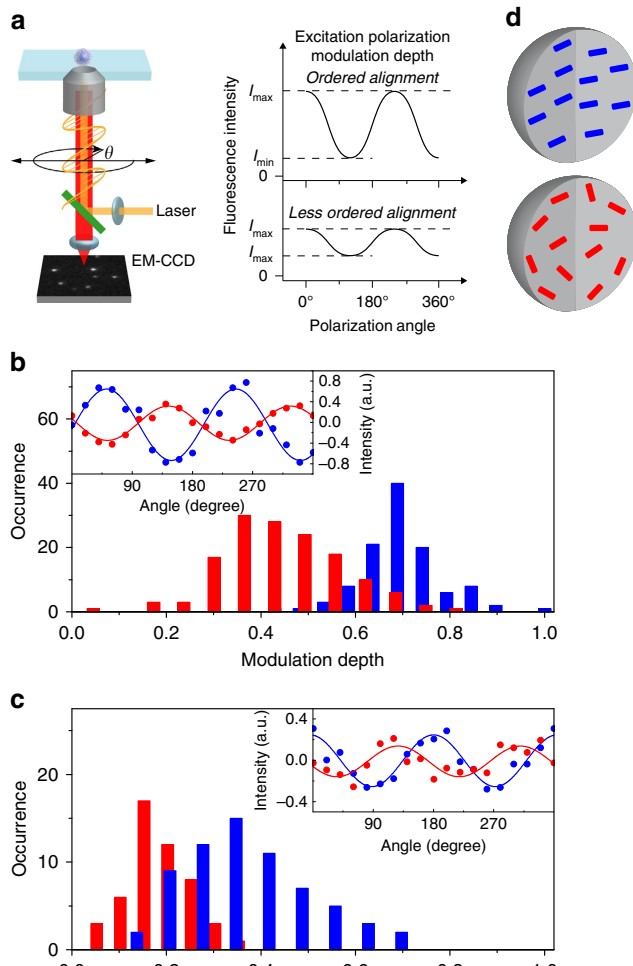

**Figure 3 | Spatial orientation of the emitting sites in the individual Pdots.** (**a**) Schematic illustration of single-particle excitation polarization modulation experiment. (**b,c**) Frequency histograms of the excitation polarization modulation depths obtained from (**b**) PD1-L (blue) and PD1-H (red) and (**c**) PD2-L (blue) and PD2-H (red). The insets show typical examples of the polarization-angle-dependent fluorescence intensity obtained for single (**b**) PD1-L (blue dots) and PD1-H (red dots) and (**c**) PD2-L (blue dots) and PD2-H (red dots) particles. The solid lines show the fitting of the data with equation (7). (**d**) Schematic illustrations of the spatial orders of the emitting sites in the Pdots fabricated at 277 K (top) and 343 K (bottom). Each rectangle corresponds to a single emitting site.

a significant change in the fluorescence lifetime between PD1-L in water and PCzBT in THF was not observed (Supplementary Fig. 1, Supplementary Table 1). The small change in the fluorescence lifetime between PD1-L in water and PCzBT in THF can be understood from the twist-angle-dependent radiative deactivation and the chain interaction-induced non-radiative deactivation (see Discussion for details). Large modification to the fluorescence spectrum and lifetime is expected when strong coherent coupling between the subunits occurs[16]. Such modifications were not observed in PD1-L in water and PCzBT in THF. Our results, therefore, suggest that the contribution of the strong coupling between the spectroscopic units (that is, dimeric form of CzBT) is negligible in the Pdots fabricated in this study.

We conducted single-particle photon coincidence measurements to estimate the numbers of independently fluorescing

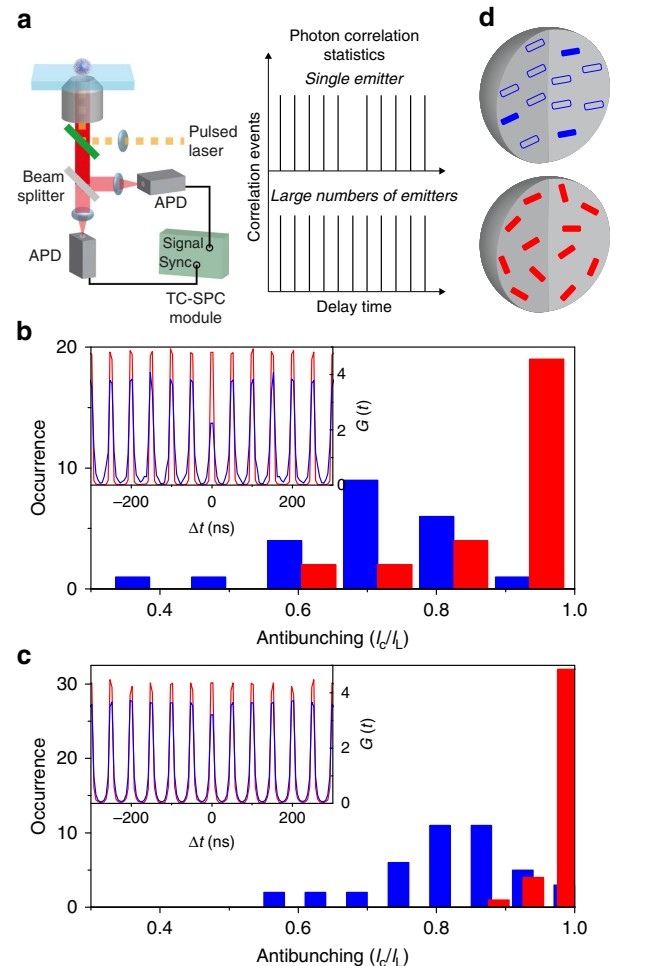

**Figure 4 | Numbers of independently fluorescing emitting sites in the individual Pdots.** (**a**) Schematic illustration of the single-particle photon coincidence experiment. (**b,c**) Frequency histograms of the $I_0/I_L$ values obtained from (**b**) PD1-L (blue) and PD1-H (red) and (**c**) PD2-L (blue) and PD2-H (red). The insets show typical examples of the photon correlation statistics obtained for single (**b**) PD1-L (blue) and PD1-H (red) and (**c**) PD2-L (blue) and PD2-H (red) particles. (**d**) Schematic illustrations of the fluorescing emitting sites in the Pdots fabricated at 277 K (top) and 343 K (bottom). The filled rectangles indicate fluorescing emitting sites. The open rectangles are the emitting sites whose excited-state energy is transferred to the lower energy sites through EET.

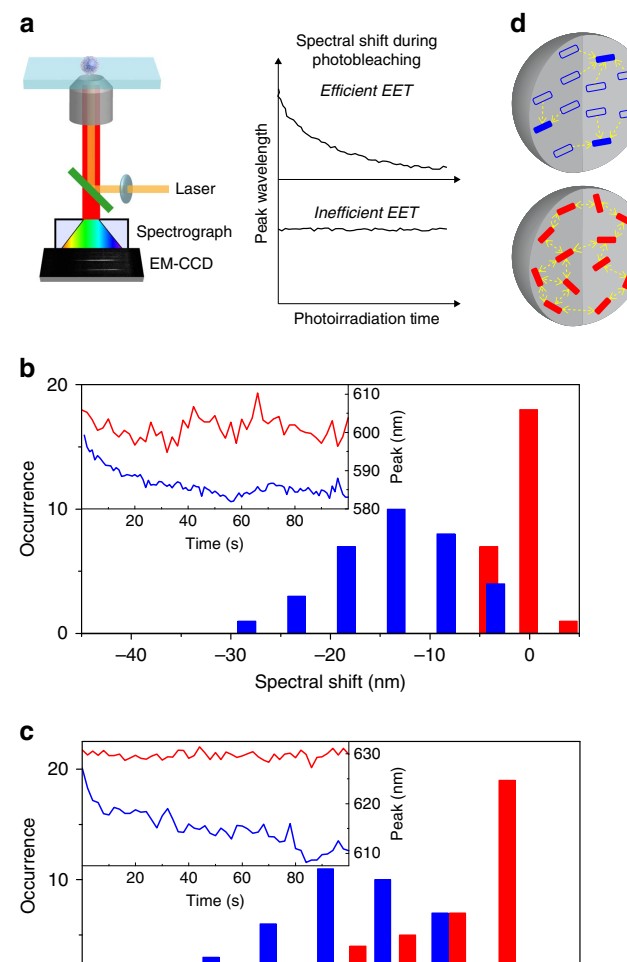

**Figure 5 | Directional EET occurring inside the individual Pdots.** (**a**) Schematic illustration of the single-particle fluorescence spectra measurement. (**b,c**) Frequency histograms of the total spectral shifts during the photobleaching obtained for (**b**) PD1-L (blue) and PD1-H (red) and (**c**) PD2-L (blue) and PD2-H (red). The insets show typical time courses of the peak emission wavelengths obtained for single (**b**) PD1-L (blue) and PD1-H (red) and (**c**) PD2-L (blue) and PD2-H (red) particles. (**d**) Schematic illustrations of the EET occurring inside the Pdots fabricated at 277 K (top) and 343 K (bottom). The arrows indicate the direction of the EET between the emitting sites.

emitting sites in each Pdot (Fig. 4a)[16]. The number of independent emitters is estimated from the intensity ratio of the peaks at the delay time of 0 ns ($I_C$) to the neighbouring peaks ($I_L$). While $I_C/I_L = 1$ is expected for multichromophoric systems with large numbers of independently emitting fluorophores, smaller $I_C/I_L$ values are obtained from multichromophoric systems with small numbers of independent emitters due to efficient singlet–singlet annihilation[30,31]. The mean $I_C/I_L$ values obtained from PD1-L and PD2-L were 0.70 and 0.82, respectively (Fig. 4b,c). These values correspond to 3.3 and 5.5 independent emitters in the PD1-L and PD2-L particles, respectively (Fig. 4d). Since each PD1-L and PD2-L contains more than 100 spectroscopic units (see Supplementary Note 1), the result indicates the high efficiency of intraparticle directional EET. In contrast, any noticeable anti-bunching behaviour was not observed in PD1-H and PD2-H (Fig. 3b,c). Since the fabricated Pdots are similar in size (Fig. 2e,h) and consequently

contain a similar number of the emitting sites, the mean $I_C/I_L$ values obtained from PD1-H and PD2-H ($I_0/I_L = 0.90$ and 0.98) indicate inefficient directional EET occurring in these particles despite their small size.

The difference in the EET efficiency was further demonstrated by the spectral shift in fluorescence during photobleaching (Fig. 5a). Blue shifts of the fluorescence peaks of PD1-L and PD2-L were observed (mean shift of 12 and 21 nm, respectively) (Fig. 5b,c, Supplementary Fig. 10a). Large blue shifts in the fluorescence spectra of multichromophoric systems are an indication of an efficient directional EET to the lowest energy site[32]. Thus, the result together with photon coincidence data strongly suggest the highly efficient funnelling of the excited state to the lowest emitting sites through EET (Fig. 5d top). In contrast, a small spectral blue shift (mean shift of 1 and 5 nm) was observed in PD1-H and PD2-H (Fig. 5b,c,

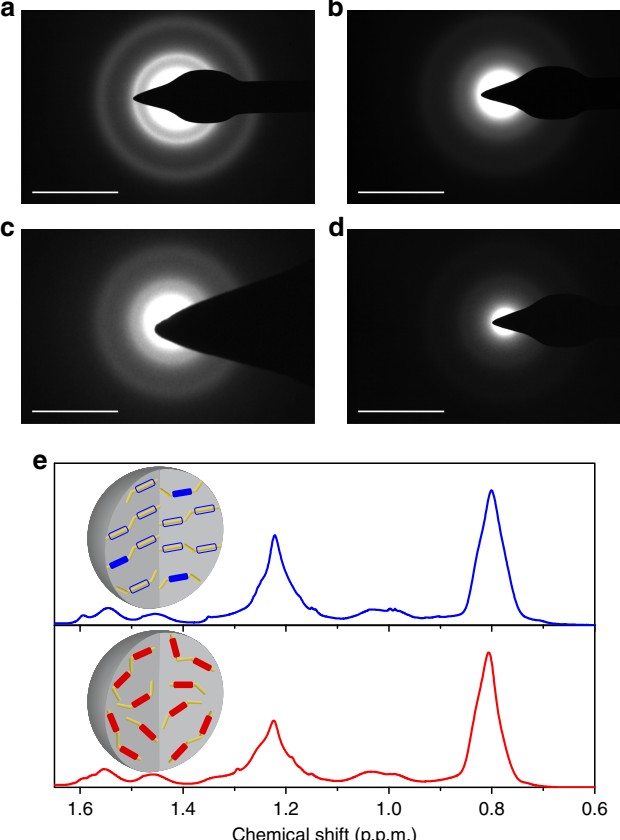

**Figure 6 | Intraparticle chain packing of the Pdots.** (**a–d**) Selected area electron diffraction patterns obtained from (**a**) PD1-L, (**b**) PD1-H, (**c**) PD2-L and (**d**) PD2-H. Scale bars = 10 nm$^{-1}$ (**e**) $^1$H NMR spectra of the PD1-L (blue line) and PD1-H (red line). The insets show schematic illustrations of the packing of the polymer chains in the Pdots fabricated at 277 K (top) and 343 K (bottom). The yellow lines signify polymer chains in the particles.

Supplementary Fig. 10b), confirming inefficient directional EET occurring in these particles (Fig. 5d bottom). The single-particle fluorescence spectroscopy experiments unambiguously demonstrated that the intraparticle EET is regulated by the spatial order of the emitting sites within the particle, even at a length scale as small as three nanometers.

**Packing of the polymer chains in the Pdots.** Next, we investigated how the spatial order of the emitting sites in the particle is established using electron microscopy. Selected area electron diffraction patterns obtained from the fabricated Pdots are shown in Fig. 6a–d. Sharp spots or thin ring structures, which indicate the presence of crystalline structures, are not observed in all Pdots, suggesting amorphous structure of the CP chains in the particles. However, the selected area electron diffraction patterns of PD1-L and PD2-L displayed more recognizable diffuse ring patterns (Fig. 6a,c) than those of PD1-H and PD2-H (Fig. 6b,d), suggesting that the chains in PD1-L and PD2-L are packed in a more structurally ordered manner than they are in PD1-H and PD2-H (inset to Fig. 6e). Since the emitting sites are a part of the main chain of the CPs, the spatially ordered arrangement of the emitting sites in PD1-L and PD2-L is a direct result of the ordered packing of the PCzBT and PCzDTBT chains in the particles.

We then used nuclear magnetic resonance (NMR) spectroscopy to investigate the packing of the aliphatic side chains in the Pdots. NMR spectra of polymers usually exhibit broad peaks because the conformational flexibility of the chains provides varied local environments surrounding specific atoms in the chain and thus varied shielding effects in each chain. Thus, the widths and shifts of the peaks provide an indication of the packing and order in the polymer chains[33]. The main peaks of $^1$H NMR spectra obtained from PD1-L and PD1-H (Fig. 6e) are located around 0.80 and 1.2 p.p.m., which are attributed to aliphatic side chains of PCzBT. A slight broadening of the peak was observed at 1.2 p.p.m. in PD1-H when compared with PD1-L. We also observed a slight spectral shift at 0.80 p.p.m. between PD1-L and PD1-H. These results can be interpreted by the different packing of the aliphatic side chains in PD1-L and in PD1-H. Given the spatially ordered packing of the main chains in PD1-L compared with those in PD1-H (Fig. 6a,b), the different packing of the aliphatic side chains could be governed by the different packing of the main chains in PD1-L and PD1-H, and vice versa.

## Discussion

Our findings provide an important clue to the development of ultrasmall Pdots. While the packing of the PCzBT chains is significantly different in PD1-L and PD1-H (Fig. 6), these two Pdots have similar sizes (Fig. 2e). This indicates that the size of the particle is determined by the PCzBT molecule itself rather than by the interaction of the PCzBT chains inside the particles. Most CPs have a linear shape, and they therefore have to adopt strained conformations if these chains are squeezed into ultrasmall Pdots[23]. In contrast, PCzBT and PCzDTBT have twisted shapes, which can be easily squeezed into ultrasmall Pdots without introducing energetically unfavourable strained conformations. This allows for the reliable and efficient fabrication of the 3-nm diameter Pdots with very high colloidal stability. Reduced surface energy of the CP chains in a bent conformation inside the Pdots might also contribute to the high stability of the particles[34].

Our findings also provide an important clue to the development of brighter Pdots. The brightness of the fluorescence is highly related to the fluorescence quantum yield ($\phi_{fl}$). $\phi_{fl}$ is determined by the ratio of the radiative rate constant ($k_r$) and the non-radiative rate constant ($k_{nr}$) (Supplementary Fig. 11a). Thus, larger $k_r$ and smaller $k_{nr}$ are required for achieving a high $\phi_{fl}$ value. Most far-red-emitting Pdots suffer from a significant reduction in $\phi_{fl}$ compared with the CPs from which they were fabricated. This is mainly due to the significant increase in $k_{nr}$ due to the generation of energy trap sites in the particles through aggregations of CP chains. Indeed, it has been widely believed that higher spatial order and better packing of CPs inside Pdots lead to interchain interactions through π–π stacking, which causes fluorescence quenching. The standard method to resolve this problem is to fabricate Pdots by mixing CPs and non-CPs[3,26] and/or inserting bulky side groups[35] to CPs, which allow for physical separation of the CPs in the Pdots[36]. While this strategy works well for planar-shaped CPs, large $\phi_{fl}$ is obtained at the cost of a lower density of fluorescing moieties[26]. Our comprehensive characterization of the Pdots fabricated using structurally similar but different CPs demonstrates that quenching can largely be suppressed without physical isolation of the CP chains by fabricating Pdots with non-planar CT-type CPs. PD1-L and PD2-L, which were fabricated without any non-CPs and/or bulky side chains, showed marginal increases of $k_{nr}$ compared with the corresponding CPs (PCzBT and PCzDTBT; Supplementary Fig. 11b,c). The generation of energy trap sites is thus inefficient

in PD1-L and PD2-L. Obviously, this is one of the reasons that relatively high $\phi_{fl}$ are maintained in PD1-L and PD2-L.

While spatially ordered packing of the chains (that is, PD1-L and PD2-L) resulted in efficient intraparticle EET, it did not cause reduction in $\phi_{fl}$, suggesting that fluorescence quenching is governed by local chain interactions rather than by the overall spatial order of the emitting sites within the particles. The main difference between most previous Pdots and PD1-L/PD2-L is the shape of the CPs. While most far-red-emitting CPs have linear and planar shapes, PCzBT and PCzDTBT have twisted nonplanar shapes. Since $\pi$–$\pi$ stacking of the CP chains is responsible for the aggregations of the chains and therefore the generation of quenching sites (that is, energy trap sites), our findings suggest that the unique twisted shape of the CPs used in this study prevents the CP chains from $\pi$–$\pi$ stacking and from forming quenching sites, which results in brighter fluorescence.

The drop in $\phi_{fl}$ in PD1-H and PD2-H compared with PD1-L and PD2-L is partly explained by the increase in $k_{nr}$ values (Supplementary Fig. 11b,c) and therefore by the generation of additional quenching sites (that is, energy trap sites) in PD1-H and PD2-H. Structural characterization (Fig. 6) revealed temperature-dependent polymer chain packing during fabrication, which demonstrated the critical role played by chain packing in the generation of quenching sites in the particles. These results clearly indicate that the brightness of Pdots is partly governed by the packing of CP chains in the particles, which can be optimized by reprecipitation conditions.

Steady-state absorption spectra of PCzBT (Fig. 1c) exhibited the largest CT absorption in the solution phase, followed by PD1-L. PD1-H exhibited smallest CT absorption. Similar trend was observed with PCzDTBT. Since fluorescence brightness is proportional to the absorption cross-section, the results indicate that PD1-L and PD2-L exhibit brighter fluorescence compared with PD1-H and PD2-H. The cross-section (which is proportional to the oscillator strength, $f$) of the CT absorption between Cz and BT depends on the rotational twist angle between the two moieties (that is, larger $f$ for smaller twist angle)[19]. Thus, the Cz and BT moieties adopt most planar conformation in THF and a more twisted conformation in PD1-H. Given the different packing of the PCzBT chains in PD1-L and PD1-H, the rotational twist angle between Cz and BT is regulated by the packing of the chains in the Pdots that is controlled by the reprecipitation conditions. According to the Strickler–Berg equation[37], $f$ is proportional to $k_r$; the rotational twist thus affects $\phi_{fl}$ as well (that is, larger $k_r$ results in larger $\phi_{fl}$) (Supplementary Fig. 11a). As predicted by the Strickler–Berg equation, the CT absorption cross-sections of PCzBT and PCzDTBT are proportional to $k_r$ (Supplementary Fig. 11b,c) except for a small deviation in PD1-H. The brighter fluorescence observed in PD1-L and PD2-L is, therefore, partly attributed to the smaller rotational twist between the Cz and BT or DTBT moieties compared with PD1-H and PD2-H. The fabrication temperature-dependent rotational twist angle is most likely due to the different CP chains packing in the Pdots.

All these observations point to a new strategy for obtaining brighter Pdots: design non-planar CPs that prevent the formation of local quenching sites and control the packing of the CP chains in Pdots by optimizing the reprecipitation conditions. EET between different CPs (that is, energy donor CP and energy acceptor CP) in single Pdots has been used to design and fabricate far-red/near-infrared emitting Pdots[3,26,38]. Although very bright Pdots can be obtained by this approach, residual fluorescence from the energy-donor CP cannot be avoided completely. Also, heterogeneity between individual particles has to be evaluated carefully in such Pdots. This will be a particularly

important issue in the fabrication of ultrasmall Pdots. Since the particles consist of a relatively small number of CP chains, the efficiency of intra-particle EET could vary between individual particles. Advanced fluorescence imaging such as fluorescence lifetime imaging and single-particle tracking with such Pdots might not be straightforward. Our new strategy to design and fabricate bright far-red/near-infrared-emitting Pdots would serve as a fundamental framework for the future development of Pdots.

We also demonstrated that EET occurring in Pdots can be regulated by controlling the physical interactions between the chains by modifying the reprecipitation conditions. Intra-particle EET is especially important in the sensing applications of Pdots[5–7,15]. While simultaneous realization of bright fluorescence with high EET efficiency is required in sensing applications, efficient EET often results in quenched fluorescence due to intrachain interactions. Our findings suggest that very bright Pdots with efficient intraparticle EET can be fabricated by selecting proper CT-type CPs and appropriate reprecipitation conditions.

While the number of independent emitters in the particles depends on the reprecipitation conditions, multiple simultaneously emitting sites exist in all the fabricated Pdots that create spatially isotropic fluorescence emitted by the particles (Supplementary Fig. 12). Spatially isotropic fluorescence is essential for accurate localization of the particles[39], which is not always straightforward with fluorophores that emit anisotropic luminescence such as organic dyes and QDs. The result together with the small size and high photostability of the obtained Pdots suggests that they could be ideal fluorophores for single-particle tracking studies[40].

Given an increasing number of applications of Pdots based on CT-type CPs because of their wide tunability of fluorescence wavelength, our findings provide a fundamental framework for designing and fabricating new functionalized Pdots. Finally, the reliable method to fabricate Pdots with 3-nm size reported here should open the possibility of using Pdots in a wide spectrum of *in vivo* applications in clinical settings.

## Methods

**Preparation of the Pdots.** Procedures for the synthesis of poly(1,8-carbazole)-benzothiadiazole copolymer (PCzBT, $M_n = 5,400$, $M_w/M_n = 1.6$) and poly(1,8-carbazole)-dithienylbenzothiadiazole copolymer (PCzDTBT, $M_n = 2,800$, $M_w/M_n = 1.1$) are described elsewhere[18]. The molecular weights of the polymers were determined by gel permeation chromatography using polystyrene standards. Similar molecular weights were obtained from matrix assisted laser desorption/ionization—time of flight (MALDI-TOF) mass spectrometry. Semiconductor QDs, Qdot 605 ITK amino (PEG) and Qtracker 655 vascular labels were purchased from Molecular Probes. THF was purchased from Sigma-Aldrich. Conjugated polymer nanoparticles (Pdots) were fabricated by a modified reprecipitation method. A stock solution was prepared by dissolving 1 mg of PCzBT in 3 ml of fresh THF. The stock solution was diluted into THF at a concentration of 1.65 p.p.m. ($3.1 \times 10^{-7}$ M), briefly sonicated in an ultrasonic bath (37 kHz, Elma Schmidbauer GmbH, Elmasonic P60H), and filtered through a 0.2-µm membrane filter (Advantec MFS., 13HP020AN). One millilitre of the solution was then added to 6 ml of MilliQ water (Millipore, Milli-Q Reference) using a syringe equipped with a 0.1-µm membrane filter (Chemglass Life Sciences, CLS-2005-003) and a 23G × 1″ needle. The solution was rapidly injected into the water in a glass vial, and the mixed solution was kept under sonication in an ultrasonic bath at either 277 or 343 K for 60 min. THF in the mixed solution was removed by evaporation under vacuum 30 min after the injection of the solution, which was followed by a continuous flow of compressed dry air for 2 h to evaporate the water and concentrate the colloidal suspension. The final volume of the colloidal solution was 200–250 µl. The resulting Pdot suspension was further filtered through a 0.1-µm membrane filter (Chemglass Life Sciences, CLS-2005-003). The Pdots fabricated in 4–6 different batches were mixed before we conducted all the experiments, including ensemble spectroscopic measurements, single-particle fluorescence microscopy experiments, electron microscopy experiments and NMR spectroscopy experiments.

**Structural characterization of the Pdots.** The zeta potential of the Pdots was measured on a dynamic light scattering Malvern Zetasizer NanoZS. TEM images of

the Pdots were produced by an FEI Tecnai G2 Spirit TWIN microscope. The samples for the TEM measurements were prepared by dropcasting colloidal solutions of the Pdots onto a holey carbon film on Cu grids (EMS, Q225-CMA) and drying them in air for 1 h immediately before the measurement. The TEM grids were treated with plasma before sample preparation. Cryo-electron microscopy (cryo-EM) images were made on an FEI Titan Krios. The samples for the cryo-EM measurements were prepared by placing 2 µl of the Pdot solution on a holey carbon coated Cu grid, quickly blotting the excess solution with filter paper to produce a thin fluid layer ($\sim$100 nm thickness), and freezing the grids in liquid ethane cryogen. The samples were then placed in the autoloader of the Titan Krios for imaging. The 1D $^1$H gradient NMR with water suppression using excitation sculpting were conducted on a Bruker Avance III 950 MHz NMR spectrometer equipped with a TCI cryo-probe. The Pdot colloidal solutions were concentrated by evaporating water using a flow of dry air through a 0.1-µm syringe membrane filter. A 4 ml of deuterated oxide was added to a 1 ml of the Pdots solutions. The solutions were concentration to the final volume of 1 ml by evaporation under vacuum. The 1D $^1$H diffusion NMR experiments using stimulated echo and bipolar gradient pulses and 3–9–19 water suppression sequence measurements to remove sharp signal impurities were performed on a Bruker Avance III 600 MHz NMR spectrometer equipped with a z-gradient broadband BBO probe.

**Ensemble spectroscopy measurements.** Steady-state absorption and fluorescence measurements were performed in a U-3900 Spectrometer (Hitachi High-Technologies) to collect absorption spectra and in a Fluoromax-4 spectrofluorometer (Horiba Scientific) to collect fluorescence spectra. After subtracting the scattering response from the solvent, fluorescence spectra were corrected using the spectral response of the photodetector. The concentration of the samples was set to $10^{-6}$ M for all fluorescence measurements to minimize inner-filter effects. The fluorescence quantum yield ($\phi_{fl}$) was determined using Rhodamine 101 (Acros Organics) and 4-(dicyanomethylene)-2-methyl-6-(4-dimethylaminostyryl)-4H-pyran (DCM, Sigma Aldrich) as references. Fluorescence decays were measured using a home-built time-correlated single-photon counting (TCSPC) setup. Samples were excited at 530 nm using an LDH-P-FA-530L picosecond pulsed diode laser (PicoQuant, pulse width $\approx$ 100 ps, repetition = 20 MHz). Fluorescence from the samples was focused on a monochromator (Sciencetech, 9030DS) to obtain spectral separation and detected with a microchannel plate photomultiplier tube (MCP-PMT, Hamamatsu Photonics K.K., R3809U-63) in the photon counting mode. The signal from MCP-PMT was amplified by a high speed amplifier (Hamamatsu Photonics K.K., C5594-12). The TCSPC measurements were conducted using a TCSPC module (PicoQuant, HydraHarp 400). The setup was controlled by SymPhoTime64 software (PicoQuant). The fluorescence decay curves were fitted to multi-exponential decaying functions with deconvolution of the instrument response function using FluoFit software (PicoQuant). Mean fluorescence lifetimes ($\tau_{fl}$) were calculated by the equation

$$\tau_{fl} = \frac{\sum_i A_i \cdot \tau_i^2}{\sum_i A_i \cdot \tau_i} \tag{1}$$

where $A_i$ and $\tau_i$ respectively denote the amplitude and lifetime of the component $i$.

**Single-particle confocal microscopy measurements.** Single-particle fluorescence measurements were conducted either on a home-built scanning confocal microscope or a wide-field epi illumination microscope, both based on an Olympus IX 71 platform. A picosecond pulsed laser diode operating at 530 nm (PicoQuant, LDH-P-FA-530L) was used as the excitation source in the scanning confocal microscope setup. The circularly polarized excitation light was obtained using a Berek compensator (Newport, Model 5540), and the laser intensity was tuned by neutral density filters (Thorlabs, FW2AND). The excitation light was expanded by a beam expander to fill the back aperture of the objective lens before introducing the light into the microscope. The samples for all the single-particle measurements were prepared by depositing colloidal solutions of the Pdots on clean coverslips. The samples were excited through a high numerical aperture (NA) oil immersion objective (UPlanSApo, ×100, NA 1.40, Olympus). The excitation power at the sample plane was set to 1.5 and 15 kW cm$^{-2}$ for the measurements of fluorescence intensity trajectories and fluorescence spectra, respectively. Raster scanning of the samples was achieved by moving the objective with a precision piezoelectric translation stage (Physik Instrumente, P-733.2CL,) controlled by a digital piezo controller (Physik Instrumente, E-710.4CL). Fluorescence from the samples was collected using the same objective, passed through a dichroic mirror (Semrock, Di02-R532) and separated from the remaining excitation light by a long-pass emission filter (Semrock, BLP01-532R) after focusing on a 100-µm pinhole (Thorlabs, P100S) by the tube lens of the microscope. Fluorescence spectra were measured by an electron-multiplying charge-coupled device (EMCCD) camera (Andor Technology, Newton 970P-BV) equipped with a spectrograph (Andor Technology, SR-303i-B). For the measurements of single-particle fluorescence images, fluorescence intensity time trajectories and fluorescence decay curves, the fluorescence signal was detected by a single photon avalanche photodiode (SPAD, PicoQuant, τ-SPAD-50,) and recorded by the TCSPC module (PicoQuant, HydraHarp 400) in the time-tagged time-resolved mode. The brightness of the fluorescence of single particles was estimated by measuring

fluorescence images of the individual particles under identical excitation and detection conditions and calculating the integrated intensity of each fluorescence spot.

**Photon coincidence measurements.** A 530-nm circularly polarized pulsed laser light (PicoQuant, LDH-P-FA-530L; 120 W cm$^{-2}$ at the sample plane) was used as the excitation source. Hanbury–Brown and Twiss-type photon correlation configurations were applied. The fluorescence photons of all wavelengths were divided by a 50/50 non-polarizing cube beam splitter (Thorlabs, CM1-BS013) and detected by two SPADs (PicoQuant, τ-SPAD-50,) after they pass through a set of bandpass filters (Semrock, FF01-665/150). The signals from the two SPADs were recorded by the TCSPC HydraHarp 400 module in the time-tagged time-resolved mode. SymPhoTime64 software (PicoQuant) was used for both the acquisition of all fluorescence microscopy data and analysis of the data.

**Determination of molar extinction coefficient of Pdots.** Molar extinction coefficients ($\varepsilon$) of the Pdots were calculated based on the peak absorbance of the ensemble absorption spectra ($A$) and their molar concentrations estimated by fluorescence correlation spectroscopy (FCS) measurements. FCS can provide concentration values when the size and shape of the confocal volume is known. A 530-nm circularly polarized pulsed laser light (PicoQuant, LDH-P-FA-530L; 1.5 kW cm$^{-2}$ at the sample plane) was used for the FCS measurements. The confocal volume ($V_{eff}$) described by the short ($r_0$) and long ($z_0$) axes of an ellipsoid ($V_{eff} = \pi^{3/2} \cdot r_0^2 \cdot z_0$) was determined by fitting the autocorrelation curve obtained from a calibrating sample, TetraSpec microspheres (0.1 µm, Molecular Probes), with a known concentration ($C$) and diffusion constant ($D = 0.044 \times 10^{-6}$ cm$^2$ s$^{-1}$ in water) to the following equation:

$$G(\tau) = X_t(\tau) \cdot \frac{1}{V_{eff} \cdot C} \cdot \frac{1}{(1 + \tau/\tau_D)} \cdot \frac{1}{\sqrt{1 + (r_0/z_0)^2 \cdot (\tau/\tau_D)}} \tag{2}$$

$$\tau_D = r_0^2 / 4D \tag{3}$$

$$N = V_{eff} \cdot C \tag{4}$$

where $\tau$, $\tau_D$, and $N$ denote the lag time, diffusion time and average particle number in the confocal volume, respectively. $X_t(\tau)$ is the contribution of triplet blinking to the autocorrelation curve and is described by

$$X_t(\tau) = 1 - T + T \cdot \exp\left(-\tau/\tau_{triplet}\right) \tag{5}$$

where $T$ and $\tau_{triplet}$ are respectively the fraction of the triplet state and the triplet lifetime. The concentration of the Pdots was determined from the amplitude of the autocorrelation curve ($1/(V_{eff} \cdot C)$) at $\tau = 0$ by fitting the data with equation (1). $C$ was determined for Pdot solutions with three different dilutions, and the $\varepsilon$ value was calculated from $A$ using the Beer–Lambert law.

**Determination of photobleaching yields.** Photobleaching yields ($\phi_{bl}$) of the Pdots were estimated by comparison with a reference sample[41]. An ensemble sample of Rhodamine 6G (R6G) dispersed in a thin film of poly(methyl methacrylate) ($\phi_{bl}^{R6G} = 5 \times 10^{-7}$)[42] was used as the reference. Fluorescence intensity trajectories of individual Pdots deposited on a cover slip and the R6G sample were measured under the same conditions using a 532-nm circulary polarized continuous wave laser light (Cobolt, Samba; 15 kW cm$^{-2}$ at the sample plane). The intensity trajectories were fitted to double exponential decaying functions. Rate constants of the photobleaching ($k_{bl}$) were determined from the fast decaying component as their contribution was dominant. $\phi_{bl}$ was estimated by the following equation:

$$\phi_{bl}^{Pdots} = \frac{k_{fl}^{R6G} \cdot k_{bl}^{Pdots}}{k_{fl}^{Pdots} \cdot k_{bl}^{R6G}} \cdot \phi_{bl}^{R6G} \tag{6}$$

where $k_{fl}$ is the rate constant of the excited-state deactivation determined by the fluorescence lifetime measurements.

**Single-particle wide-field microscopy measurements.** A continuous wave diode laser module operating at 488 nm (Cobolt, MLD 488) was used as the excitation source. The laser intensity was tuned with a variable neutral density filter (Thorlabs, RSP1D). A circularly polarized excitation light was obtained using a Berek compensator (Newport, Model 5540). Samples were excited through an oil immersion objective (Olympus, TIRF UApo N, ×100, NA 1.49) by focusing the excitation light at the back aperture of the objective lens. Resulting fluorescence was separated from the excitation laser by a dichroic beam splitter (Semrock, FF506-Di03) followed by an emission filter (Semrock, FF01-609/181) and collected by a high-speed EMCCD camera (iXon Ultra 897, Andor Technology).

**Excitation polarization modulation measurements.** For the excitation polarization modulation measurements, a linearly polarized excitation beam was obtained by inserting a Glan-laser polarizer (Thorlabs, GL5-A) into the circularly polarized excitation beam (Cobolt, MLD 488). The linearly polarized excitation

beam was rotated around the optical axis by using a half-wave plate (Thorlabs, WPMQ05M-488) mounted on a precision rotation mount (Thorlabs, RS1D) while consecutive fluorescence images of the samples were recorded. The excitation power at the sample plane was set to $120\,W\,cm^{-2}$. The fluorescence intensities of individual particles were calculated by integrating the intensities of each pixel in the fluorescence spots. The fluorescence intensity at polarization angle $\theta$ ($I_\theta$) was fit to the following equation:

$$I_\theta = I_0[1 + M \cdot \cos\{2(\theta - \theta_0)\}] \qquad (7)$$

where $M$, $I_0$ and $\theta_0$ denote the excitation polarization modulation depth, average fluorescence intensity and polarization angle at maximum fluorescence intensity, respectively. Alternatively, $M$ was calculated by the following equation:

$$M = (I_{max} - I_{min})/(I_{max} + I_{min}) \qquad (8)$$

where $I_{max}$ and $I_{min}$ are the maximum and minimum fluorescence intensities, respectively.

**Defocused fluorescence imaging.** Defocused fluorescence imaging was performed by using the circularly polarized excitation laser beam (Cobolt, MLD 488). The excitation power at the sample plane was set to $500\,W\,cm^{-2}$. The fluorescence images were recorded by shifting the microscope stage by approximately 1 μm towards the objective from the focal position. The fluorescence images were magnified 2.14 times by inserting a relay lens (Edmund Optics, 43-995) before the EMCCD camera. The defocused images were recorded at a 0.5-Hz frame rate.

**Data availability.** The data that support the findings of this study are available from the corresponding author on reasonable request.

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

## Acknowledgements

The research reported in this publication was supported by King Abdullah University of Science and Technology (KAUST). We thank Dr Rachid Sougrat for the cryo-EM images and Dr Christian Canlas and Dr Abdelhamid Emwas for the NMR spectra of the fabricated Pdots. We also thank Haruka Osako for the assistance in the polymer synthesis.

## Author contributions

S.H. conceived the project. H.P. and S.H. designed the experiments. H.P. conducted the experiments and analysed the data. T.M. synthesized the polymers. H.P. and S.H. wrote the manuscript. All authors discussed the results.

## Additional information

**Competing interests:** The authors declare no competing financial interests.

