## [Peer Review File · Nature Communications]

Reviewers' comments:

Reviewer #1 (Remarks to the Author):

The authors demonstrate exceptionally bright fluorescent polymer dots based on two copolymers, PCzBT and PCzDTBT. In both cases it was found that the polymer dots at higher temperatures were less fluorescent, and less ordered, based on NMR line width analysis, polarization modulation measurements and electron diffraction measurements. EET was evaluated using photon coincidence measurements and the results found more efficient EET at lower temperatures. The authors attributed the decrease in fluorescence at higher temperatures to be due to an increase in the D/A torsional angle. They maintain that the twisted copolymer dots are better able to pack without inducing fluorescence quenching.

Overall, the paper may be interesting to a specialists in the field of biological imaging. The fundamental science as implied in the abstract is, however, somewhat lacking. They fall short of showing a conclusive relationship between chain interactions and photophysical properties. The photon correlation measurements presented in fig. 3 are particularly difficult to decipher but even if EET was accurately measured its not clear that difference at low and higher temperatures are due to chain packing and in particular to increased intrachain twisting. The paper is publishable but not in Nature Communications.

Reviewer #2 (Remarks to the Author):

This manuscript describes a novel approach to producing fluorescent organic nanoparticles with potential applications as probes in biological systems. The small (several nanometer size) fluorescent nanoparticles are based on donor-acceptor type conjugated polymers with non-planar main chain backbone conformations that the authors suggest prevents the formation of quenching interactions that are present in particles formed from planar polymers. The effects of nanoparticle formation temperature on final emission properties are investigated and differences interpreted in terms of the efficiency of intraparticle energy transfer. A variety of experimental techniques including single particle fluorescence spectroscopy have been applied to support the conclusions of the paper. The outcomes provide a new paradigm for making very small fluorescent nanoparticles that I expect to be of considerable interest to the scientific community. The paper could have been strengthened with an example of the uptake of the new nanoparticles in biological samples to illustrate their advantages over other types of fluorescent probes. Other matters that would improve the manuscript are highlighted below.

(i) an important aspect is the ability to reproduce nanoparticle synthesis. The procedure outlined in the methods section (page 12) describes the addition of a dilute THF solution of the polymer into water with sonication "over 60 minutes". The authors should clarify if the addition is made rapidly then sonicated for 60 minutes, or if the additions is made during the 60 minute time-frame while also sonicating. Were the average particle diameters and standard deviations quoted from one batch or from multiple batches? How reproducible was the particle formation process.

(ii) The authors indicate that there was no "fluorescence blinking" in the intensity-time trajectories of the nanoparticles (main text and Figure S6). What was the bin-time resolution for these measurements. Are the authors suggesting that there is no triplet state formation in the polymers and no electron transfer processes?

(iii) A minor grammatical correction is that I suggest the authors change the phrase "with peaking at xxx nm..." on page 4, lines 16 and 21 to "with maxima at xxx nm,.."

Reviewer #3 (Remarks to the Author):

The manuscript makes an important contribution that goes beyond the current understanding of the photophysical properties of conjugated polymer nanoparticles (CPNs). In addressing very small particles (< 5 nm), at the same time a very relevant size regime is touched upon. As CPNs are studied widely, amongst others for bioimaging purposes, and this is a broadly interdisciplinary topic, the results and interpretation reported and concepts delineated should appeal to the readership of Nature Communications.

A point of concern are the molecular weights of the polymers employed. The PCzDTBT possesses an Mn of 2800 g/mol according to the experimental section. This is probably vs. linear PS standards, meaning the true molecular weight is even a factor of two to three lower. This corresponds to only very few repeat units, rather than true chains. This will impact the conformations and fluorescence properties enormously. Furthermore, there could even be a significant compositional heterogeneity (1,8-carbazol vs. dithenylbenzothiadiazole repeat units). The authors should comment on this. Concerning EET processes, relevant work by Schwartz should also be cited (e.g. Annu. Rev. Phys. Chem. 2003, 54, 141) on page 3 and beyond.

On page 5, the authors state that the formation of ultras small particle sizes is due to the twisted shapes of the polymers studied, allowing them to be squeezed in a small volume without introducing an energetically unfavorable strained conformation. It has been shown from studies of conformations of single rigid chains in nanoparticles, that the minimization of surface energy actually overrides the energy required for chain bending (Phys. Chem. Chem. Phys. 2015, 17, 32289), which should be cited here. Further, the low molecular weights of the polymers need to be considered here. How long would an extended chain actually be compared to the particle size?

Page 6, it appears contradictive that the reprecipitation protocol should not affect particle size and morphology, while the internal structure is altered. Perhaps this is a question of the interpretation of the term 'morphology'. Is 'shape' more appropriate here?

Preparation procedures of the nanoparticles are obviously important for their properties. This topic has been reviewed comprehensively ('Nanoparticles of Conjugated Polymers' in Chem. Rev. 2010, 110, 6260).

Page 6, 7th line of the 2nd paragraph. Please check labelling (PD2-L rather than PD2-H?).

The NMR spectra of the two different polymers' particles (page 9, Figure 4) do not allow for conclusions on differences (of conformational packing) in the particles. Given the very different quality of the two spectra in terms of signal to noise, small differences in peak width do not appear significant.

Point-by-point response to the referees

Reviewer 1

1. *Overall, the paper may be interesting to a specialists in the field of biological imaging.*

Thank you for the positive comment on our work.

2. *The fundamental science as implied in the abstract is, however, somewhat lacking. They fall short of showing a conclusive relationship between chain interactions and photophysical properties. The photon correlation measurements presented in fig. 3 are particularly difficult to decipher*

As mentioned in the main text, the photon correlation measurement is a powerful method to determine the number of independently emitting fluorophores in multichromophoric systems and has been used to characterize EET between the fluorophores in many multichromophoric systems (e.g. ref. 30, 31). The method has also been used to quantify the intra- and inter-chain EET occurring in the conjugated polymers (ref. 16). These previous studies have concluded that the anti-bunching behavior (i.e., small I_C/I_L values) is the direct consequence of EET. In this study, we designed the experimental setup, analyzed the obtained data, and interpreted the data according to these previous studies. We believe that we report the accurate data with appropriate interpretation. In order for readers to easily understand this part of the manuscript, in the revised manuscript, we added a schematic illustration highlighting the relationship between the photon correlation data and the EET occurring inside the Pdots. We also cited the relevant reference paper (ref. 16) in the revised manuscript to help readers to better understand the concept and the interpretation of the photon correlation experiments.

3. *even if EET was accurately measured its not clear that difference at low and higher temperatures are due to chain packing and in particular to increased intrachain twisting.*

While we discussed both overall chain packing and rotational twist of the chain within each emitting site in the original manuscript, the discussion might not be very clear. In the following, we describe the two points separately.

The overall packing of the chains inside the Pdots was discussed based on the selected-area electron diffraction (SAED) images and NMR spectra obtained from the Pdots fabricated at the two different temperatures (Fig. 6). These experiments clearly demonstrated the better overall chain packing in the Pdots fabricated at the low temperature (PD1-L and PD2-L). Since the emitting sites are a part of the main chain of the CPs, the ordered packing of the PCzBT and PCzDTBT chains in the particles is a direct cause of the spatially ordered arrangement of the emitting sites in PD1-L and PD2-L that we observed in Fig. 3b and

3c. Clearly, the overall packing of the CP chains affect significantly the spatial orders of the emitting sites and thus the EET inside the particles (Figs. 4b, 4c, 5b, 5c).

The fluorescence lifetime experiments also suggest the critical role played by the chain packing on the brightness of the fluorescence obtained from the Pdots. The fluorescence quantum yield (ϕ_f) is determined by the ratio of the radiative rate constant (k_r) and the non-radiative rate constant (k_{nr}) (Fig. S11a). In most far-red-emitting Pdots, the efficient π - π stacking of the CP chains inside the particles results in the generation of energy trap sites to which the efficient EET occurs, leading to the significant increase in k_{nr} (i.e., significant reduction of fluorescence brightness). Our results clearly showed marginal increases of k_{nr} in PD1-L and PD2-L compared with the corresponding CPs (PCzBT and PCzDTBT; Fig. S11b, S11c), which is in contrast to the much larger k_{nr} observed in PD1-H and PD2-H. The results suggest that the CP chains in the Pdots fabricated at the two different temperatures interact in very different way. Therefore, the temperature dependent packing of the CP chains demonstrated in Fig. 6 indicates that the k_{nr} and thus the fluorescence brightness is at least partly governed by the overall packing of the chains.

Our experiments also demonstrated that while spatially ordered packing of the chains (i.e., PD1-L and PD2-L) resulted in efficient intraparticle EET (Figs. 3, 4, 5), it did not cause significant increase in k_{nr} , suggesting that fluorescence quenching is governed by local chain interactions rather than by the overall spatial order of the emitting sites in the particles. The main difference between most previous Pdots and PD1-L/PD2-L is the shape of the CPs. While most far-red-emitting CPs have linear/planar shapes, PCzBT and PCzDTBT have nonlinear/nonplanar shapes. Since π - π stacking of the CP chains is responsible for the aggregations of the chains and therefore the generation of quenching sites (i.e., energy trap sites), our findings suggest that the unique nonplanar shape of the CPs used in this study prevents the CP chains from π - π stacking and from forming quenching sites, which results in brighter fluorescence.

The second point was probably not very clear in the original manuscript. In the following, the involvement of the rotational twist of the chains within each emitting site in the fluorescence brightness is described based on the steady-state absorption spectra (Fig. 1c and 1d) and the radiative rate constant (k_r) (Fig. S11b and S11c) obtained by the fluorescence lifetime measurements (Fig. S1). Steady-state absorption spectra of PCzBT (Fig. 1c) exhibited the largest charge transfer (CT) absorption in the solution phase, followed by PD1-L. PD1-H exhibited smallest CT absorption. Similar trend was observed with PCzDTBT. Since fluorescence brightness is proportional to the absorption cross-section, the results indicate that PD1-L and PD2-L exhibit brighter fluorescence compared with PD1-H and PD2-H. The cross-section (which is proportional to the oscillator strength, f) of the CT absorption between Cz and BT depends on the rotational twist angle between the two moieties (i.e., larger f for smaller twist angle) (reported in ref. 19). Thus, the Cz and BT moieties adopt most planar conformation in THF and a more twisted conformation in PD1-H. Given the different packing of the PCzBT chains in PD1-L and PD1-H, the rotational twist angle between Cz and BT is regulated by the packing of the chains in the Pdots that is controlled by the reprecipitation conditions. According to the Strickler-Berg equation, f is proportional to k_r ; the rotational twist thus affects ϕ_f as well (i.e., larger k_r results in larger ϕ_f) (Fig. S11a). As predicted by

the Strickler-Berg equation, the CT absorption cross-sections of PCzBT and PCzDTBT are proportional to k_r (Fig. S11b, S11c) except for a small deviation in PD1-H. The brighter fluorescence observed in PD1-L and PD2-L is, therefore, partly attributed to the smaller rotational twist between the Cz and BT or DTBT moieties compared with PD1-H and PD2-H. The fabrication temperature-dependent rotational twist angle is most likely due to the different CP chains packing in the Pdots.

All these observations point to a completely new strategy for obtaining brighter Pdots: design non-planar CPs that prevent the formation of local quenching sites and control the packing of the CP chains in Pdots by optimizing the reprecipitation conditions. We added the above discussion to the revised manuscript.

Reviewer 2

1. *The outcomes provide a new paradigm for making very small fluorescent nanoparticles that I expect to be of considerable interest to the scientific community.*

Thank you for the positive comment on our work.

2. *The paper could have been strengthened with an example of the uptake of the new nanoparticles in biological samples to illustrate their advantages over other types of fluorescent probes.*

We agree with the reviewer and fully recognize the importance of testing our Pdots in bioimaging studies. Indeed, we are planning to insert carboxyl groups to PCzBT and PCzDTBT for conjugating our Pdots to biological macromolecules such as proteins to evaluate the performance of our Pdots in fluorescence imaging experiments. The main focus of this study, however, is to develop new strategy to design and fabricate ultrasmall Pdots with bright fluorescence and evaluate fundamental fluorescence properties of the fabricated Pdots. We will report the applicability of our Pdots to bioimaging experiments in a separate paper in near future.

3. *an important aspect is the ability to reproduce nanoparticle synthesis. The procedure outlined in the methods section (page 12) describes the addition of a dilute THF solution of the polymer into water with sonication "over 60 minutes". The authors should clarify if the addition is made rapidly then sonicated for 60 minutes, or if the additions is made during the 60 minute time-frame while also sonicating. Were the average particle diameters and standard deviations quoted from one batch or from multiple batches? How reproducible was the particle formation process.*

We fully agree with the reviewer's comment. Indeed, the development of the reliable and reproducibile

protocol of the fabrication of the ultrasmall Pdots was one of the important achievements of this study although we did not emphasize this point too much in the manuscript. We confirmed that we were able to obtain the Pdots with very similar size distribution and fluorescence quantum yield when we followed the protocol developed in this study. We also confirmed that the effect of the temperature during the reprecipitation/ultrasonication on the fluorescence properties as well as chain packing inside the particles was reproducible. The average particle diameters and standard deviations were determined by using 4-6 batches (i.e., the Pdots fabricated in 4-6 different batches were mixed before we conducted all the experiments, including ensemble spectroscopic measurements, single-particle fluorescence microscopy experiments, electron microscopy experiments, and NMR spectroscopy experiments). Regarding the detailed protocol of the fabrication, the polymer solution was injected into the water by the syringe rapidly and the mixture was kept sonicated in the ultrasonic bath for 60 min. We added the detailed protocol of the fabrication in the Methods section of the revised manuscript.

4. *The authors indicate that there was no "fluorescence blinking" in the intensity-time trajectories of the nanoparticles (main text and Figure S6). What was the bin-time resolution for these measurements. Are the authors suggesting that there is no triplet state formation in the polymers and no electron transfer processes?*

We recorded the single-particle fluorescence intensity time trajectories using the time-tagged time-resolved (TTTR) mode of the time-correlated single-photon counting (TCSPC) module. This allows us to re-bin the data after the data acquisition. We show the trajectory with four different bin-times (500, 100, 50, and 10 ms) in Fig. 2-4. We didn't see any "blinking" behavior even at the 10 ms temporal resolution. Although we cannot rule out the triplet blinking occurring at much shorter time scale, this is rather unlikely because the photon correlation experiments (Figs. 4b, 4c) suggest the presence of multiple independently emitting sites inside each Pdot, even for the particles that show the efficient intraparticle EET. We added the above discussion in the revised manuscript.

Fig. 2-4. Fluorescence intensity time trajectories obtained from single Pdots. (a-d) Intensity trajectories obtained from a single PD1-L particle with the bin sizes of (a) 500 ms, (b) 100 ms, (c) 50 ms, and (d) 10 ms. (e) Intensity trajectory obtained from a single PD2-L particle.

5. *A minor grammatical correction is that I suggest the authors change the phrase "with peaking at xxx nm..." on page 4, lines 16 and 21 to "with maxima at xxx nm,..."*

Thank you for the suggestion. According to the reviewer's suggestion, we rephrased the sentences in the revised manuscript.

Reviewer 3

1. *As CPNs are studied widely, amongst others for bioimaging purposes, and this is a broadly interdisciplinary topic, the results and interpretation reported and concepts delineated should appeal to the readership of Nature Communications.*

Thank you for the positive comment on our work.

2. *A point of concern are the molecular weights of the polymers employed. The PCzDTBT possesses an M_n of 2800 g/mol according to the experimental section. This is probably vs. linear PS standards, meaning the true molecular weight is even a factor of two to three lower. This corresponds to only very few repeat units, rather than true chains.*

As the reviewer mentioned, the reported molecular weights of the CPs were determined using PS standards. To estimate the molecular weights of the CPs, we performed MALDI-TOF mass spectrometry experiments (see Fig. 3-2-1 and Fig. 3-2-2). The molecular weights estimated by the MALDI-TOF mass spectrometry experiments ($M_n \approx 5,000$ and 2,800 for PCzBT and PCzDTBT) are similar to those calculated using the PS standards ($M_n \approx 5,400$ and 2,800 for PCzBT and PCzDTBT). We mention this in the revised manuscript. The molecular weights correspond to 7.9 and 3.3 repeat units for PCzBT and PCzDTBT, respectively.

b

n	ser.	rep.unit	resid.	end1	end2	cation	Mn	Mw	pd	DP	% I.	cnt
1	1	684.422	157.234				4823.26	5382.74	1.11600	7.04721	14.8	11
2	2	684.422	22.6978				5530.59	6152.24	1.11240	8.08068	16.1	13
3	3	684.422	79.1501				4141.25	4514.45	1.09012	6.05073	2.8	8
4	4	684.422	114.228				4521.57	4926.23	1.08949	6.60641	2.1	8
5	5	684.422	211.896				3806.55	3982.18	1.04614	5.56170	0.9	5
6	6	684.422	248.657				3699.88	4023.34	1.08742	5.40585	1	6
7	7	684.422	278.941				3925.12	4326.18	1.10218	5.73494	1.6	7
8	8	684.422	291.583				3953.20	4266.18	1.07917	5.77597	6.1	7
9	9	684.422	338.537				2843.05	3121.30	1.09787	4.15395	0.8	5
10	10	684.422	362.123				6101.48	6432.41	1.05424	8.91480	2.3	8
11	11	684.422	413.066				3694.26	4226.22	1.14400	5.39764	5	8
12	12	684.422	437.503				7148.96	7575.67	1.05969	10.4453	5.6	11
13	13	684.422	496.305				5466.84	5958.08	1.08986	7.98754	3.8	11
14	14	684.422	547.882				3142.02	3715.53	1.18253	4.59077	7.5	9
15	15	684.422	572.210				6329.01	6872.92	1.08594	9.24724	11.3	13
16	16	684.422	629.746				5035.54	5466.76	1.08563	7.35737	3.9	9
17	17	684.422	664.252				5337.07	5643.71	1.05745	7.79792	1.7	8
18	18	684.422	682.027				2892.78	3358.85	1.16111	4.22661	3.8	8
19	19	684.422	529.245				6095.01	6226.84	1.02163	8.90534	0.7	5
20	20	684.764	298.720				8841.41	8932.73	1.01033	12.9116	1.6	5

Fig. 3-2-1. Estimation of the molecular weight of PCzBT. (a) MALDI-TOF mass spectrum of PCzBT. The spectrum was measured on a Bruker UltrafleXtreme mass spectrometer, which was operated in a linear-positive ion mode. *trans*-2-[3-(4-*tert*-butylphenyl)-2-methyl-2-propenylidene]malononitrile (DCTB) was used as a matrix. (b) Molecular weights calculated from the mass spectrum.

Fig. 3-2-2. Estimation of the molecular weight of PCzDTBT. (a) MALDI-TOF mass spectrum of PCzBT. The spectrum was measured on a Bruker UltrafleXtreme mass spectrometer, which was operated in a linear-positive ion mode. Dithranol was used as a matrix. (b) Molecular weights calculated from the mass spectrum.

3. *This will impact the conformations and fluorescence properties enormously. Furthermore, there could even be a significant compositional heterogeneity (1,8-carbazol vs. dithenylbenzothiadiazole repeat units). The authors should comment on this.*

As the reviewer pointed, the lengths of the CP chains used in this study are relatively short. Therefore, each PCzBT and PCzDTBT chain may adopt different conformational states, and thus may exhibit different fluorescence properties. Indeed, the single-chain experiments revealed the large heterogeneity in the fluorescence properties between the chains (ref. 19). However, the single-particle experiments showed the narrow distributions of the fluorescence brightness (Fig. S4, Table S3) and lifetime (Fig. S8, Table S3), demonstrating the homogeneous fluorescence properties of the fabricated Pdots. We note that the distribution of the brightness and lifetime obtained from our Pdots are comparable to those obtained from

commercially available quantum dots (QD605 and QD655) (Table S3). The narrow distributions are probably due to the large number of the CP chains in each Pdot (see Supplementary Note). We added the above discussion in the revised manuscript.

4. *Concerning EET processes, relevant work by Schwartz should also be cited (e.g. Annu. Rev. Phys. Chem. 2003, 54, 141) on page 3 and beyond.*

Thank you for the suggestion. According to the reviewer's suggestion, we added the reference paper in the revised manuscript.

5. *On page 5, the authors state that the formation of ultrasmall particle sizes is due to the twisted shapes of the polymers studied, allowing them to be squeezed in a small volume without introducing an energetically unfavorable strained conformation. It has been shown from studies of conformations of single rigid chains in nanoparticles, that the minimization of surface energy actually overrides the energy required for chain bending (Phys. Chem. Chem. Phys. 2015, 17, 32289), which should be cited here.*

Thank you for the valuable comment. According to the reviewer's suggestion, we added below discussion about the effect of the surface energy on the conformational state of the CP Chains inside the Pdots and added the reference paper in the revised manuscript. "Reduced surface energy of the CP chains in a bent conformation inside the Pdots might also contribute to the high stability of the particles (ref. 34)."

6. *Further, the low molecular weights of the polymers need to be considered here. How long would an extended chain actually be compared to the particle size?*

The contour lengths of the CPs are estimated to be 15.4 nm and 9.5 nm for PCzBT and PCzDTBT, respectively. Although it is not easy to determine the chain lengths (i.e., end-to-end distance) since PCzBT and PCzDTBT are not linear-chain polymers, the size of the Pdots (3-4.5 nm in diameter) is much smaller than the lengths of the CPs, suggesting that the chains are confined to the space smaller than their lengths.

7. *Page 6, it appears contradictive that the reprecipitation protocol should not affect particle size and morphology, while the internal structure is altered. Perhaps this is a question of the interpretation of the term 'morphology'. Is 'shape' more appropriate here?*

Thank you for the comment. We used the term "morphology" to describe the particle-level (i.e., not chain level) morphology, including the shape of the particles. Since it seems that the word is confusing and misleading, we replace the word with "shape" in the revised manuscript.

8. *Preparation procedures of the nanoparticles are obviously important for their properties. This topic has been reviewed comprehensively ('Nanoparticles of Conjugated Polymers' in Chem. Rev. 2010, 110, 6260).*

Thank you for the suggestion. According to the reviewer's suggestion, we added the reference paper in the revised manuscript.

9. *Page 6, 7th line of the 2nd paragraph. Please check labelling (PD2-L rather than PD2-H?).*

Thank you for pointing out the error. We corrected the error in the revised manuscript.

10. *The NMR spectra of the two different polymers' particles (page 9, Figure 4) do not allow for conclusions on differences (of conformational packing) in the particles. Given the very different quality of the two spectra in terms of signal to noise, small differences in peak width do not appear significant.*

We conducted the NMR spectroscopy experiment again to obtain spectra with better signal to noise ratio (Fig. 3-10). Our new experiment demonstrated a slight broadening of the peak at 1.22ppm (the peak width is approximately 20% broader for PD1-L compared with PD1-H). We also observed a slight spectral shift at 0.80 ppm between PD1-L and PD1-H. These results can be interpreted by the different packing of the aliphatic side chains in PD1-L and PD1-H. Since the difference between the two spectra is relatively small, we rephrased the sentences related to the NMR spectroscopy experiment and discuss the different packing of the aliphatic chains without emphasizing the structural order in the particles in the revised manuscript.

Fig. 3-10. ¹H NMR spectra obtained for PD1-L (top) and PD1-H (bottom).

REVIEWERS' COMMENTS:

Reviewer #1 (Remarks to the Author):

My main concerns were somewhat addressed in the revised manuscript. The influence of chain packing on fluorescence is a complex problem which is only partially explained by the distribution of trap sites. The way the chromophores interact within the p-stacks is also important - for example, exciton theory predicts fluorescent quenching in H-type aggregates and superradiance in Scheibe aggregates. The revised manuscript is improved but I am however neutral as to whether the paper should be published in Nature Communications.

Reviewer #2 (Remarks to the Author):

In this revised manuscript the authors have satisfactorily addressed the technical issues I had raised in my original review. While I do feel the manuscript would be stronger with an example of the performance of these fluorescent nanoparticles in a biological sample, the authors indicate in their response that this aspect will be addressed in a subsequent publication. On balance I believe there is sufficient novelty in the preparation and characterisation of these very bright fluorescent nanoparticles to have significant impact and be of interest to the broad readership of the journal.

Reviewer #3 (Remarks to the Author):

In the revised manuscript, the authors have addressed all questions from my previous review in a satisfactory way.

With clarification of the low molecular weights (3 repeat units for the PCzDTBT on average), it appears debatable whether 'chain packing' arguments apply to these molecules.

Notwithstanding, I believe this manuscript should appeal to the readership of Nature Communications

Point-by-point response to the referees

Reviewer 1

- 1. My main concerns were somewhat addressed in the revised manuscript. The influence of chain packing on fluorescence is a complex problem which is only partially explained by the distribution of trap sites. The way the chromophores interact within the p-stacks is also important - for example, exciton theory predicts fluorescent quenching in H-type aggregates and superradiance in Scheibe aggregates. The revised manuscript is improved but I am however neutral as to whether the paper should be published in Nature Communications.*

Thank you for the constructive comment on our manuscript. As the reviewer pointed out, the chain packing should have a complex effect on the fluorescence properties of the Pdots. Indeed, we cannot completely rule out the possibility of forming either H- or J (Scheibe)-type aggregates inside the particles. However, our results suggest that they play a minor role in the photophysical properties of the Pdots. The absorption spectra of the Pdots fabricated at the low and high temperatures exhibited similar spectral shape and peak position. They also agreed well with the absorption spectrum of the dimeric form of the conjugated polymer (CPs). Those results suggest that the exciton delocalization occurs in up to two monomer units along the chains (i.e., interchain interaction has a minor effect on the exciton delocalization). Also, we did not observe a significant change in the fluorescence lifetime between the Pdots and the corresponding CPs. This rules out the strong coupling between the spectroscopic units. Instead, our findings are reasonably interpreted by the twist-angle-dependent formation of quenching sites inside the particles. We continue our research on the development of new conjugated polymer nanoparticles, and we will try to address the issues raised by the reviewer through our future works.

Reviewer 2

- 1. In this revised manuscript the authors have satisfactorily addressed the technical issues I had raised in my original review. While I do feel the manuscript would be stronger with an example of the performance of these fluorescent nanoparticles in a biological sample, the authors indicate in their response that this aspect will be addressed in a subsequent publication. On balance I believe there is sufficient novelty in the preparation and characterisation of these very bright fluorescent nanoparticles to have significant impact and be of interest to the broad readership of the journal.*

Thank you for the positive comment on our manuscript. Thanks to the reviewer's comments, the manuscript has been significantly improved. We will explore the possibility of using our conjugated

polymer nanoparticles for bioimaging applications, and will try to address their capability and applicability as a fluorescent tag in our future works.

Reviewer 3

- 1. In the revised manuscript, the authors have addressed all questions from my previous review in a satisfactory way. With clarification of the low molecular weights (3 repeat units for the PCzDTBT on average), it appears debatable whether 'chain packing' arguments apply to these molecules. Notwithstanding, I believe this manuscript should appeal to the readership of Nature Communications.*

Thank you for the positive comment on our manuscript. Thanks to the reviewer's comments, the manuscript has been significantly improved. We continue our research on the development of new conjugated polymer nanoparticles, and we will try to address the issues raised by the reviewer through our future works.